# Reductive inactivation of the hemiaminal pharmacophore for resistance against tetrahydroisoquinoline antibiotics

Wan-Hong Wen[1], Yue Zhang[1], Ying-Ying Zhang[1], Qian Yu[2], Chu-Chu Jiang[2], Man-Cheng Tang[1], Jin-Yue Pu[1], Lian Wu[1], Yi-Lei Zhao [ID] [2], Ting Shi[2 ✉], Jiahai Zhou[3 ✉] & Gong-Li Tang [ID] [1,4 ✉]

Antibiotic resistance is becoming one of the major crises, among which hydrolysis reaction is widely employed by bacteria to destroy the reactive pharmacophore. Correspondingly, antibiotic producer has canonically co-evolved this approach with the biosynthetic capability for self-resistance. Here we discover a self-defense strategy featuring with reductive inactivation of hemiaminal pharmacophore by short-chain dehydrogenases/reductases (SDRs) NapW and homW, which are integrated with the naphthyridinomycin biosynthetic pathway. We determine the crystal structure of NapW·NADPH complex and propose a catalytic mechanism by molecular dynamics simulation analysis. Additionally, a similar detoxification strategy is identified in the biosynthesis of saframycin A, another member of tetra-hydroisoquinoline (THIQ) antibiotics. Remarkably, similar SDRs are widely spread in bacteria and able to inactive other THIQ members including the clinical anticancer drug, ET-743. These findings not only fill in the missing intracellular events of temporal-spatial shielding mode for cryptic self-resistance during THIQs biosynthesis, but also exhibit a sophisticated damage-control in secondary metabolism and general immunity toward this family of antibiotics.

[1] State Key Laboratory of Bio-organic and Natural Products Chemistry, Center for Excellence in Molecular Synthesis, Shanghai Institute of Organic Chemistry, University of Chinese Academy of Sciences, Chinese Academy of Sciences, 345 Lingling Road, Shanghai 200032, China. [2] State Key Laboratory of Microbial Metabolism, Joint International Research Laboratory of Metabolic and Developmental Sciences, School of Life Sciences and Biotechnology, Shanghai Jiao Tong University, Shanghai 200240, China. [3] CAS Key Laboratory of Quantitative Engineering Biology, Shenzhen Institute of Synthetic Biology, Shenzhen Institute of Advanced Technology, Chinese Academy of Sciences, Shenzhen 518055, China. [4] School of Chemistry and Material Sciences, Hangzhou Institute for Advanced Study, University of Chinese Academy of Sciences, 1 Sub-lane Xiangshan, Hangzhou 310024, China. ✉email: tshi@sjtu.edu.cn; jiahai@siat.ac.cn; gltang@sioc.ac.cn

To combat the worldwide rise of antibiotic resistance, understanding the resistance mechanism at the molecular level is paramount, which is obligatory to discover various antibiotics or directedly design more effective analogs[1–6]. Many antibiotics contain hypersensitive and highly reactive pharmacophore, which is the basis for displaying biological activity as warhead. As a result, the producing-organisms have to co-evolve an effective self-protection scheme with the biosynthetic capability dealing with the dangerous warhead to avoid injuring themselves. Among the known self-resistance strategies, enzyme-catalyzed hydrolysis is widely employed by bacteria to destroy the reactive pharmacophore in either antibiotic-resistant strain of pathogenic bacteria or antibiotic-producers. One of the well-known examples is β-lactamase-mediated hydrolytic deactivating of β-lactam antibiotics, which not only widely spread in the pathogenic and environmental microbiome but also led to the development of β-lactamase inhibitors as next generation of drugs[7,8]. Recently, different family of cyclopropane hydrolases were identified to catalyze hydrolysis of cyclopropane warhead, conferring self-resistance involved in yatakemycin (YTM)/CC-1065 and colibactin biosynthesis[9,10]. Therefore, continuous efforts on elucidation for different kinds of self-resistance mechanisms based on natural product biosynthesis will enrich our knowledge about enzyme-catalyzed inactivation of antibiotics, which may include the other enzyme-reactions acting on the pharmacophore beyond hydrolysis.

Tetrahydroisoquinoline (THIQ) antibiotics, with a special THIQ framework, have attracted continuous studies due to the complex polycyclic structures and excellent biological activities against bacteria and tumor cells[11–13]. This family of natural products includes more than 60 members exemplified by naphthyridinomycin (NDM, 1), saframycin S (SFM-S, 5), ecteinascidin 743 (ET-743, 7), lemonomycin (LMM, 9) (Fig. 1a), and so on, in which ET-743 has been approved as an anticancer drug[14]. As a family of DNA damaging genotoxins, THIQ antibiotics exhibit remarkably potent DNA alkylating activity through the departure of hydroxy group in hemiaminal moiety followed by an attack of N-2 of guanine in GC-rich region (Fig. 1b)[11]. Thereby the hemiaminal pharmacophore serves as a warhead for antibiotic and antitumor potential, which calls effective resistance mechanism to counter its toxicity by the producing microbes.

In previous biosynthetic studies of NDM[15,16], we found an extracellularly oxidative activation and conditionally over-oxidative inactivation of an intermediate 7H-NDM (2) by a secreted enzyme, NapU, controlling the concentration of NDM around cells for self-resistance, and prodrug 12 was detected during biosynthetic process (Fig. 1c)[17]. This raises an open question: how the inactivated intermediate 12 is generated in cytoplasm and which enzyme is responsible for inactivating the hemiaminal pharmacophore (Fig. 1c)? Here, we solved these mysteries by elucidating an intracellularly reductive inactivation of NDM catalyzed by NapW and homW, which belong to short-chain dehydrogenase/reductase (SDR) family encoded by genes napW within biosynthetic gene cluster (BGC) and homW outside BGC, respectively. We further demonstrated that expression of the genes endows bacteria with NDM-resistance and identified the key residue, Asp165, involved in NapW-catalytic reduction based on structural analysis. In this work, we found that similar self-resistance is involved in the biosynthesis of SFM; and this family of SDRs from other microorganisms could also recognize NDM and other three THIQ compounds to reductively inactivate the hemiaminal warhead. The substrate promiscuity of SDR proteins thereby not only provides the detoxification protection in cytoplasm during the biosynthetic pathway and general

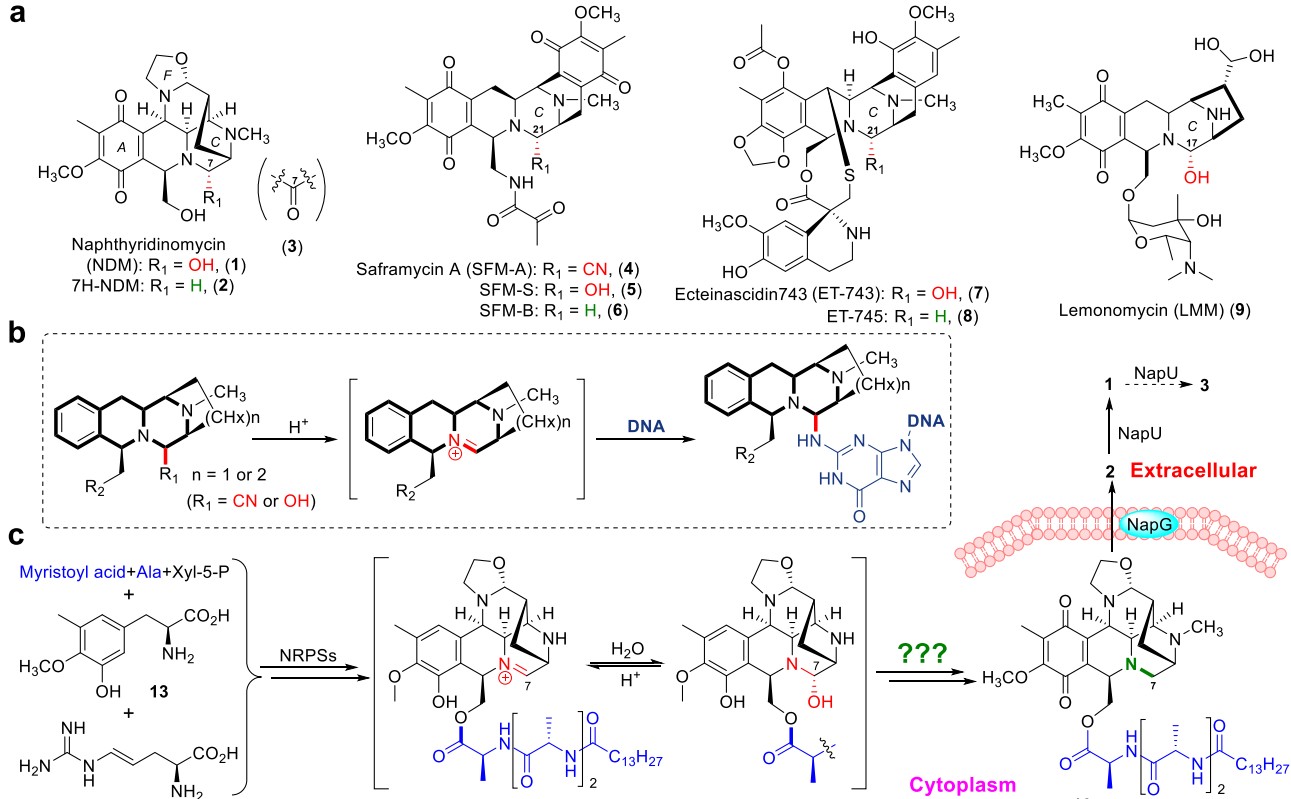

**Fig. 1 Members of THIQ antibiotics, biosynthesis, and activation of NDM. a** Chemical structures of typical THIQ antibiotics. **b** Proposed mechanism of THIQ antibiotic-mediated DNA alkylation. **c** Brief summary of NDM biosynthetic pathway featuring with intracellular pharmacophore quenching and extracellular reactivation.

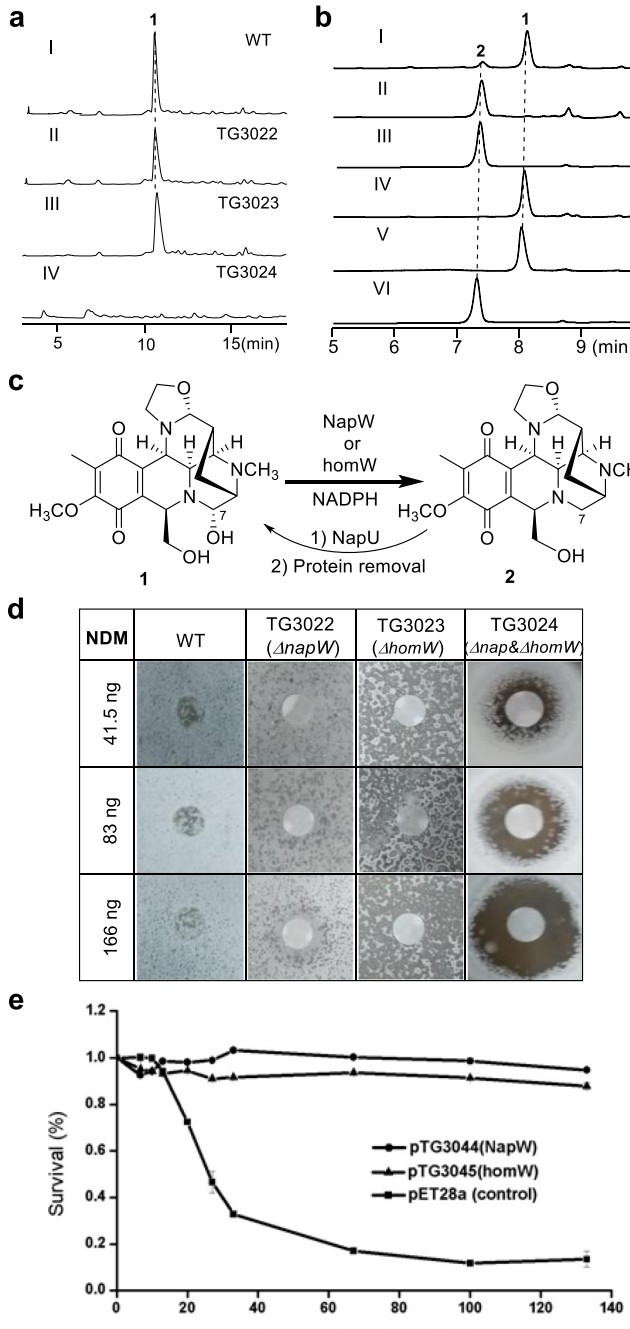

**Fig. 2 Investigation of reductases NapW and homW in NDM biosynthesis. a** Genetic characterization in vivo. HPLC analysis with UV detected at 270 nm (I) wild-type *S. lusitanus* NRRL 8034; (II) mutant *S. lusitanus* TG3022 (*ΔnapW*); (III) mutant *S. lusitanus* TG3023 (*ΔhomW*); (IV) double mutant *S. lusitanus* TG3024 (*ΔnapW&ΔhomW*). **b** HPLC analysis of the enzymatic reaction products: full reaction catalyzed by NapW for 10 min using NADH (I) or NADPH (II) as cofactor; (III), full reaction mediated by homW; (IV), control reaction with heat-inactivated NapW; (V), **1** generated in situ by NapU from **2**; (VI), standard of **2**. **c** Chemical reactions by NapU and NapW/homW. **d** NDM resistance of *S. lusitanus* WT and the mutants. **e** Comparison of the antibacterial activity of NDM against *E. coli* BL21 (DE3) containing pET28a (vector), pTG3044 (*napW*), and pTG3045 (*homW*).

immunity toward THIQ antibiotics, but also expands the known enzymatic reactions for pharmacophore modification beyond hydrolysis.

## Results

### NapW and homW catalyzed reductive inactivation of NDM for self-resistance.
The oxidative reactions performed extracellularly by the secreted oxidase NapU occur at the late stage of the biosynthetic pathway, while, the chemical logic of formation of the inactivated intermediate **12** remains unclear. One functionally unassigned SDR encoded by *napW* in the BGC, NapW, attracts our attention, because the SDR family proteins could catalyze many different reactions including reduction of C=N bond[18–20]. To explore the function of NapW, we inactivated the *napW* gene by in-frame deletion and obtained the *ΔnapW* mutant *Streptomyces lusitanus* TG3022 (Supplementary Fig. 1). Compared with the metabolite profiles of the wild type (WT), the mutant still produced **1** but no other compounds (Fig. 2a). This result suggested *napW* might not work in NDM biosynthesis or there should be other homologous gene to complete its function. Indeed, *homW* was found outside of the BGC by genome sequencing of NDM-producing bacteria, *S. lusitanus* NRRL 8034. HomW exhibits high sequence similarity (75% identity) with NapW and was speculated to play the same role as NapW. To verify the speculation, we knocked out *homW* gene in WT and TG3022 to acquire *ΔhomW* (TG3023) and *ΔnapW&ΔhomW* (TG3024) mutants, respectively (Supplementary Fig. 2). The fermentation result of *ΔhomW* is identical with that of *ΔnapW* and WT; however, the double knock-out mutant (*ΔnapW&ΔhomW*) eliminated the **1** production (Fig. 2a). These in vivo genetic evidences suggested that *napW* and its orthologue gene *homW* could functionally compensate each other, both of them are required for NDM biosynthesis.

To further demonstrate how NapW/homW mediate the inactivation of hemiaminal warhead, we next performed biochemical characterizations. Given the fact that the proposed biosynthetic intermediate **10** or **11** (Fig. 1c) is not available, we chose to employ **1** as the substrate to do the reduction assay (Fig. 2c). In a previous work, we verified that secreted flavoprotein NapU catalyzed oxidative activation of inactive compound **2** to afford toxic product **1** extracellularly, as well as over-oxidative inactivation of **1** into **3** by controlling the concentration of **1** for self-protection[17]. Since **1** still exhibits the ability to diffuse into cells, we reasoned that NapW and homW should catalyze the inactivation of **1** to yield **2** for avoiding endogenous DNA alkylation. We firstly expressed two genes in *Escherichia coli* and purified the two proteins (Supplementary Fig. 3); then carried out NapU-catalyzed reaction to acquire unstable **1**, which is difficult to isolate by fermentation, and subsequently removed NapU by ultrafiltration (Fig. 2c). When the purified recombinant NapW or homW was incubated with **1** in the presence of NADPH, the expected reduced compound **2** is effectively accumulated along with the disappearance of **1** (Fig. 2b–II, III). When the cofactor NADPH is replaced by NADH, the enzymatic efficiency was dramatically decreased (Fig. 2b–I). These results strongly demonstrate that NapW and homW functionally catalyze the reductive inactivation of the highly reactive iminium species, which implied that the two SDR proteins may participate in self-protection from endogenous DNA alkylation.

To verify this assumption, we examined the relationship between these two genes and the self-resistance of the producer against NDM (**1**). Compared with the WT and single gene inactivation strain *S. lusitanus* TG3022 (*ΔnapW*) or TG3023 (*ΔhomW*), the double-deletion strain *S. lusitanus* TG3024 (*ΔnapW&ΔhomW*) exhibited obviously increased sensitivity to NDM (Fig. 2d), suggesting both *napW* and *homW* as NDM resistant determinants. Therefore, the expression of *napW* or *homW* can protect *S. lusitanus* from cytotoxic effect when NDM is biosynthesized. We next used *E. coli* BL21 (DE3) cells as test

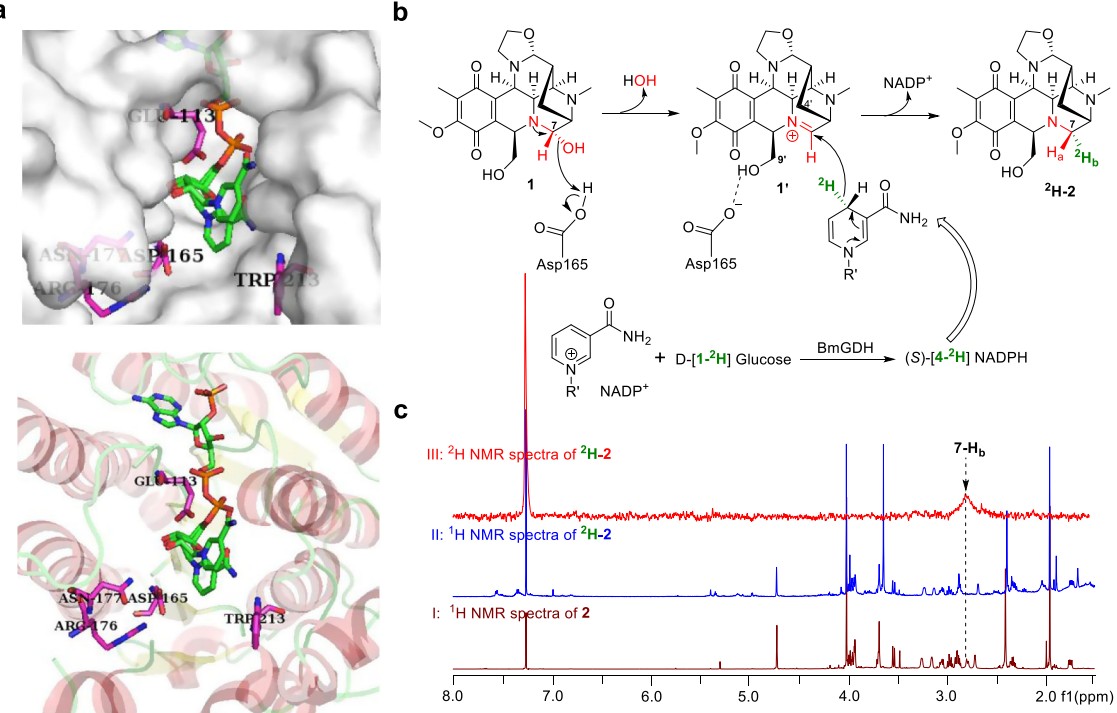

**Fig. 3 Enzymology characterization of NapW-catalyzed reduction in NDM biosynthesis. a** Crystal structure of NapW-NADP$^+$ complex. In this cartoon model, polar amino acids (Glu113, Asp165, Arg176, Asn177, and Trp213) around nicotinamide terminal shown as sticks. **b** Proposed enzymatic mechanism. **c** NMR analysis of enzymatic products.

strain to estimate the resistance to NDM by overexpression of *napW* or *homW* in vivo. The survival ratio suggested that the recombinant strain is obviously resistant to NDM (Fig. 2e). Collectively, these results unambiguously supported our proposal that the SDRs NapW and homW axiomatically endow cells resistance to NDM for self-protection.

**Structure-based enzymology of NapW-catalyzed reaction.** To probe the molecular basis of the NapW/homW-mediated self-protection, we purified the NapW protein and determined its X-ray crystal structure at 2.1 Å resolution by molecular replacement method using the structure of human SDR family member one (PDB ID: 2QQ5) as searching model. The structure of NapW contains a typical Rossmann fold with seven β sheets embraced by α helices (Supplementary Fig. 4), which is the key feature of SDR family proteins for binding cofactors[21]. We further obtained the structure of NapW·NADP$^+$ binary complex by co-crystallizing NapW, NADP(H) and compound **1** or **2** (Fig. 3a). Although the electron density of compound **1** or **2** was missing, the nicotinamide moiety of NADP$^+$ shows conformational isomerism in the binary complex structure (Supplementary Fig. 5). The binding of cofactor NADP$^+$ enables loop 47–58 and loop 107–119, which are too flexible to observe in apo-NapW structure, adopt stable conformation and are clearly observed. The surface of this complex reveals the adenine terminus of NADP$^+$ is deeply buried inside NapW and the nicotinamide moiety which delivers hydrogen to substrate locates in a wide pocket on the surface (Fig. 3a). We mutated several polar residues around the nicotinamide moiety to alanine and found only mutant D165A showed obviously decreased catalytic activity (Supplementary Figs. 6, 7). The result indicated that Asp165 may participate in the catalytic process.

To better investigate the substrate binding mechanism and the role of Asp165, we constructed the tertiary complex model of NapW·NADPH·1 by molecular docking and carried out molecular dynamics (MD) simulation experiments (Supplementary

Figs. 8–10). We found that Asp165 could form a hydrogen bond with the C-7 hydroxy of **1** to stabilize the substrate and facilitate the dehydration process. Therefore, the distance between O3 of **1** and carboxyl of Asp165 was analyzed in four times 50 ns MD trajectories. The average distance was 3.91 Å, indicating the stable hydrogen bond existed indeed. To verify the function of residues Asp165, MD simulations were also performed for D165A mutant in NapW·NADPH·1 (Fig. 4a). In the D165A mutant system, substrate **1** escaped from the catalytic center due to the loss of traction from Asp165. To be precise, two key distances representing the critical reaction steps (proton transfer and hydride delivery) were selected to describe these conformational changes. The former was the distance between O3 of **1** and Cβ(CB) of Asp165, and the latter was that between C7 of **1** and C4N of NADPH. Compared to the two-dimensional distance scatter plot in WT, the distance distribution in the D165A mutant looked more incompact, suggesting that **1** kept away from the catalytic center in D165A mutant (Fig. 4a). From these observations, we verified that Asp165 played a critical role in stabilizing the substrate in the catalytic site.

Next, we investigated the catalytic mechanism of NapW with the quantum mechanical/molecular mechanical (QM/MM) method. As shown in Fig. 3b, it was proposed that Asp165 could facilitate the transfer of proton from the carboxyl to the C-7 hydroxy, resulting in dehydration to form the electrophilic iminium species **1′**, and then the hydride delivery from NADPH to **1′** could generate the product **2**. To obtain the relative stable conformations for QM/MM calculations, two MD trajectories were extended to 150 ns for sufficient sampling, and we chose the representative structure from the major cluster of the two trajectories as the initial structure to gain the reaction energy profile. Moreover, we found there was always water around O3 of **1** (Supplementary Fig. 11). Therefore, the computational model, as shown in Fig. 4b, consisted of NADPH, substrate **1**, Asp165, a water molecule, and other residues within 5 Å of NADPH and **1**.

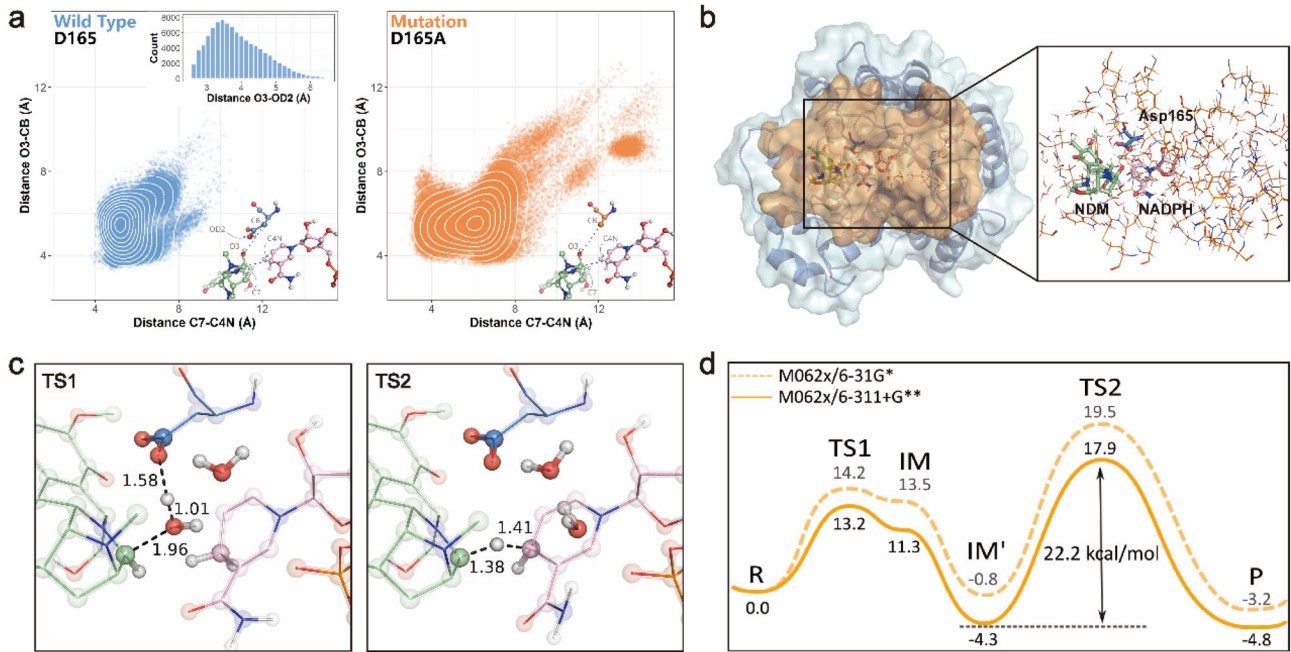

**Fig. 4 The MD simulations and QM/MM calculations for NapW/NADPH-NDM model. a** The two-dimensional scatter plot of distance C7-C4N and O3-CB. The wild-type system was shown in blue and the mutation system was shown in orange. In each model subplot, the structure of active site including NDM (green), NADPH (pink), and residue 165 (blue or orange) was shown in the bottom right. The important atoms and distances were annotated in gray. The distance O3-OD2 was counted and shown in the wild in the upper right in the wild-type system subplot. **b** The computational model for QM/MM calculations. The protein NapW was shown in blue while the orange part was represented the computational model. The QM region was shown in the form of a ball and stick model on the right. The part shown in the line form was set in MM region. **c** The optimized structures of key transition states (TSs). The key distances related to the catalytic reaction of NapW were labeled. **d** The reaction energy profile for NapW/NADPH-NDM model.

In the first step of dehydration, Asp165 protonated the hydroxyl of NDM to produce a water molecule with the energy barrier of 13.2 kcal/mol (Fig. 4c, d). After that, NADPH provided a proton to the intermediate **1′** from the same direction as that of losing hydroxyl of **1**, generating the product **2** with the energy barrier of 22.2 kcal/mol. As we can see, the second reaction step is rate-limiting in the overall catalytic process of NapW.

To further explore the mechanism, NapW-catalyzed reaction was conducted in the presence of (S)-[4-$^2$H] NADPH which is obtained by glucose dehydrogenase, BmGDH from *Bacillus megaterium* DSM 2894 with D-[1-$^2$H] Glucose as the deuterium donor and NADP$^+$ as cofactor[22,23]. 1 Da increase at $m/z$ of $^2$H-**2** was detected in NapW/BmGDH cascade reaction (Supplementary Fig. 12), which indicated the deuterium was transferred to **1′** (Fig. 3b). Then, large-scale reactions were performed to isolate $^2$H-**2** for elucidating the exact direction of deuterium incorporation at C-7 position. $^1$H–$^1$H NOESY spectrum of **2** shows that the NOE signal (indicated by red dashed arrows) between 4′-H$_a$ and 7-H$_a$, distinguishing the positions of 7-H$_a$ (2.91 ppm) and H$_b$ (2.80 ppm) (Supplementary Fig. 13). The $^1$H-NMR spectra of $^2$H-**2** indicated the peak disappearance of 7-H$_b$ when compared with that of compound **2** from end to end. Expectedly, the $^2$H NMR spectrum of $^2$H-**2** shows an obvious peak at 2.80 ppm (Fig. 3c). In conclusion, the direction of hydride supplied by NADPH is identical to that of hydroxyl leaving, which illustrates the reductive reaction is an S$_N$1 process, but not an S$_N$2 reaction. This is in line with our computational results. Owing to the block of C-9′ and C-4′ groups, the hydride tends to attack **1′** from the direction of a smaller steric hindrance (Fig. 3b).

**SDRs-catalyzed inactivation of hemiaminal-bearing prodrug in SFM-A biosynthesis**. After confirming the self-resistant role of

NapW/homW in NDM biosynthetic pathway, we wonder whether the similar reductive inactivation of hemiaminal warhead exists in other THIQ antibiotics producing bacteria. We have been pursuing the biosynthetic studies of SFM-A (**4**, Fig. 1a)[24–27]; therefore, the *sfm* BGC and the genome of SFM-A producer, *S. lavendulae* NRRL 11002, were carefully analyzed to search for genes encoding NapW-like SDRs. Indeed, in addition to SfmO1 (56% identity with NapW) encoded by *sfmO1* within *sfm* BGC, there are two another homologous proteins homO1a (62% identity) and homO1b (60% identity) encoded by genes outside the BGC. We then expressed and purified these three proteins from *E. coli* for biochemical assay (Supplementary Fig. 14). Following the proposed model of NDM (Fig. 1c), we predicted that the propeptide iminium intermediate **16** (Fig. 5a) should presumably be the true substrate of SfmO1 (or homO1a/homO1b). Once **15/16** is generated by nonribosomal peptide synthetase (NRPS), this hemiaminal pharmacophore-containing intermediate will be immediately reduced into inactive compound **17** catalyzed by SfmO1 and homO1a/homO1b in case of its potent toxicity to host DNA (Fig. 5a). Given the fact that SfmC-mediated enzymatic synthesis of **15** was established by Oikawa's group[28], we thereby followed this method to perform the dual Pictet-Spengler reaction catalyzed by SfmC using **13** and **14** as substrates (Fig. 5a). MS analysis showed the enzymatic product was indeed identical with **16** containing the iminium (Fig. 5c, HR-MS/MS in Supplementary Fig. 15). When the SDRs and NADPH were further added into this assay, we observed the generation of reductive product **17** (Fig. 5b), which was further confirmed by HR-MS/MS (Fig. 5d and Supplementary Fig. 16). Though homO1b shows lower efficiency, SfmO1, and homO1a exhibit similar activity. To our surprise, NapW and homW also could effectively catalyze the same reaction (Fig. 5b). These collective evidences not only support our proposed self-resistance model in

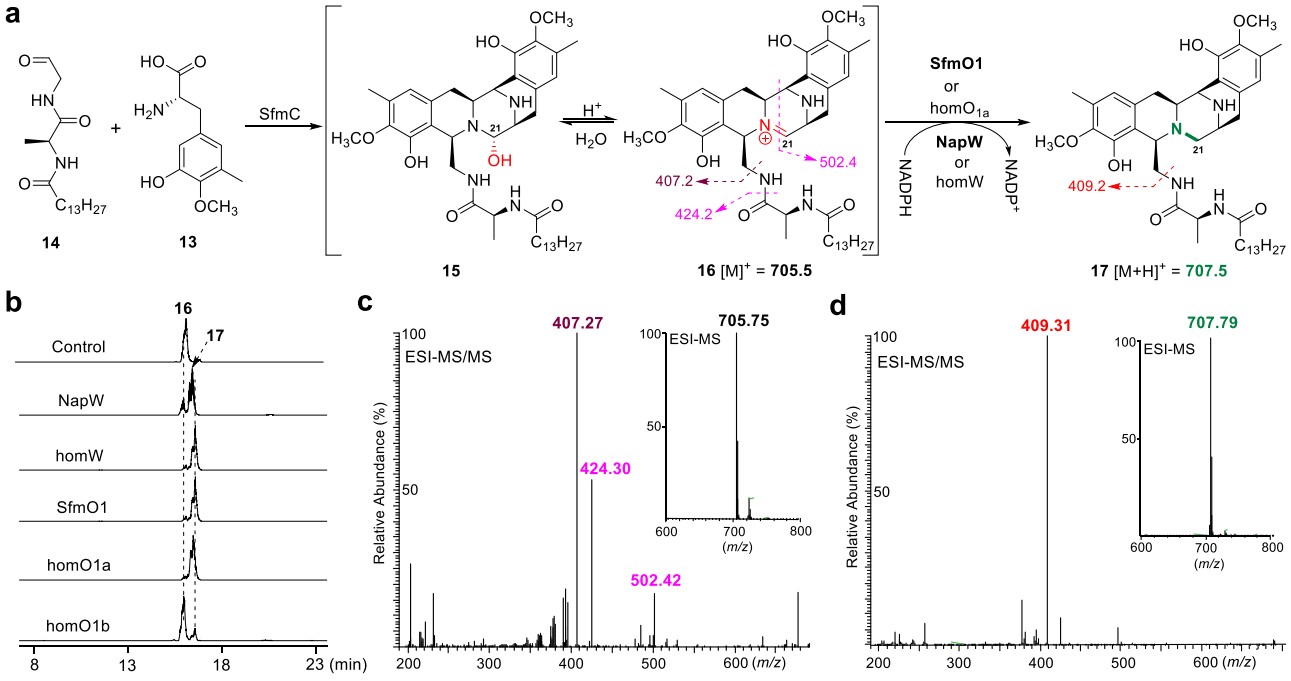

**Fig. 5 Biochemical investigation of SDRs-catalyzed pharmacophore reductive quenching in SFM biosynthesis. a** Enzymatic reactions mediated by SfmC and SDRs. **b** LC-MS analysis of enzymatic reaction products. **c** ESI-MS and ESI-MS/MS analysis of **16** generated by SfmC. **d** ESI-MS and ESI-MS/MS analysis of **17** generated by SDRs.

NDM and SFM-A biosynthesis, but also suggest that this family of SDRs may exhibit broad substrate-tolerance.

**SDRs from antibiotic-producing microbes catalyzed the inactivation of other THIQ antibiotics.** Considering these SDRs show substrate promiscuity to reductively quench the common hemiaminal pharmacophore, we thereby tested more THIQ antibiotics including ET-743 (**7**), a clinically approved anticancer drug discovered from marine tunicate *Ecteinascidia turbinate*[14,29]. Surprisingly, each of the five SDRs originated from NDM or SFM-A producer could effectively catalyze the reductive inactivation of **7** to yield a warhead-blocked derivate **8** (Fig. 6a), which were shown by HPLC analysis (Fig. 6b) and confirmed by MS analysis (Fig. 6c, d; HR-MS/MS in Supplementary Fig. 17). Moreover, we tried to obtain LMM (**9**) by fermentation of *S. candidus* LL-AP191[30]. However, we isolated an analog, 16-dehydroxy-LMM (**18**, Fig. 6e), which was characterized by NMR analysis (Supplementary Fig. 18 and Supplementary Table 4). Using **18** as substrate, we found that NapW, homW, SfmO1, and homO1a exhibit catalytic activity but homO1b does not (Fig. 6f). Large-scale enzymatic reactions were performed and the reductive product **19** was isolated, the structure of which was confirmed by NMR (Supplementary Fig. 19 and Supplementary Table 5) and demonstrated to lack a hydroxy group at C-17 (Fig. 6e). Furthermore, a small quantity of **19** could be detected in fermentation broth of LMM-producing bacteria by HPLC-MS (Supplementary Fig. 20), which strongly supported that its genome encoded one or more NapW-like proteins for self-resistance. These results demonstrated that the SDRs originated from THIQ antibiotic-producing microbes indeed exhibit recognition promiscuity toward diverse substrate harboring THIQ backbone with hemiaminal moiety.

**SDRs from non-antibiotic-producing microbes catalyze inactivation of THIQ antibiotics.** The genes outside the BGC also encode the isozymes exhibiting the same physiological function, which hints similar isozymes may exist in other microbes. Indeed, homologous genes of *napW*/*homW* and *sfmO1*/*homO1a*/*homO1b*

are widely distributed in nature, especially microbiome genome. All 4771 sequences (above 34% sequence identity with NapW) were clustered by sequence similarity network (SSN) analysis (Supplementary Fig. 21), which show that they were from different microorganisms including eukaryotic and human microbes with actinobacteria as major sources. Within the genome of *S. lividans* 1326, a non-THIQs producing actinobacteria, two SDR proteins bearing high homology with NapW: SlvW1 (74% identity) and SlvW2 (61% identity) were found. Besides, we randomly selected several proteins with different identities (75–34% identity) from different microorganisms, including SDR-Pb (*Paenibacillus*, Firmicutes), SDR-Cc (*Chroococcales cyanobacterium*, Cyanobacteria), SDR-Rm (*Rhizobium mesoamericanum*, Proteobacteria), SDR-Mb (*Myxococcales bacterium*, Proteobacteria), SDR-Li (*Leptospira interrogans*, Spirochaetes), SDR-Cs (*Chlorella sorokiniana*, Eukaryota), and SDR-Ss (*Sporomusa silvacetica*, Firmicutes) (Supplementary Table 6), to obtain the recombinant proteins from *E. coli* BL21 (DE3) for enzymatic essays toward previously mentioned four THIQs (Supplementary Fig. 22). HPLC analyses showed that most proteins can catalyze reductive reaction of THIQ compounds except that SDR-Li and SDR-Ss exhibited significantly lower activity for **1**, **16**, and **7** and no activity for **18** (Supplementary Figs. 23–26). This result suggested the promiscuous substrate recognition of NapW-homologs may endow their hosts with potentially general resistance against THIQ antibiotics physiologically, even for the non-antibiotic-producing microbes.

**Discussion**

Combining genetic characterization and biological investigation, we now filled in the missing intracellular events of a multi-dimensional temporal-spatial shielding mode for the self-resistance during THIQs biosynthesis in *Streptomyces*. The BGC encodes a major facilitator subfamily transporter presumably responsible for the antibiotic effluxes[16,24,31], which was assumed as the first action conferring self-resistance. In addition, an UV-repair protein was considered as the second defense for the producer via nucleotide excision-based DNA repair[16,24,32]. In fact, both of the canonical

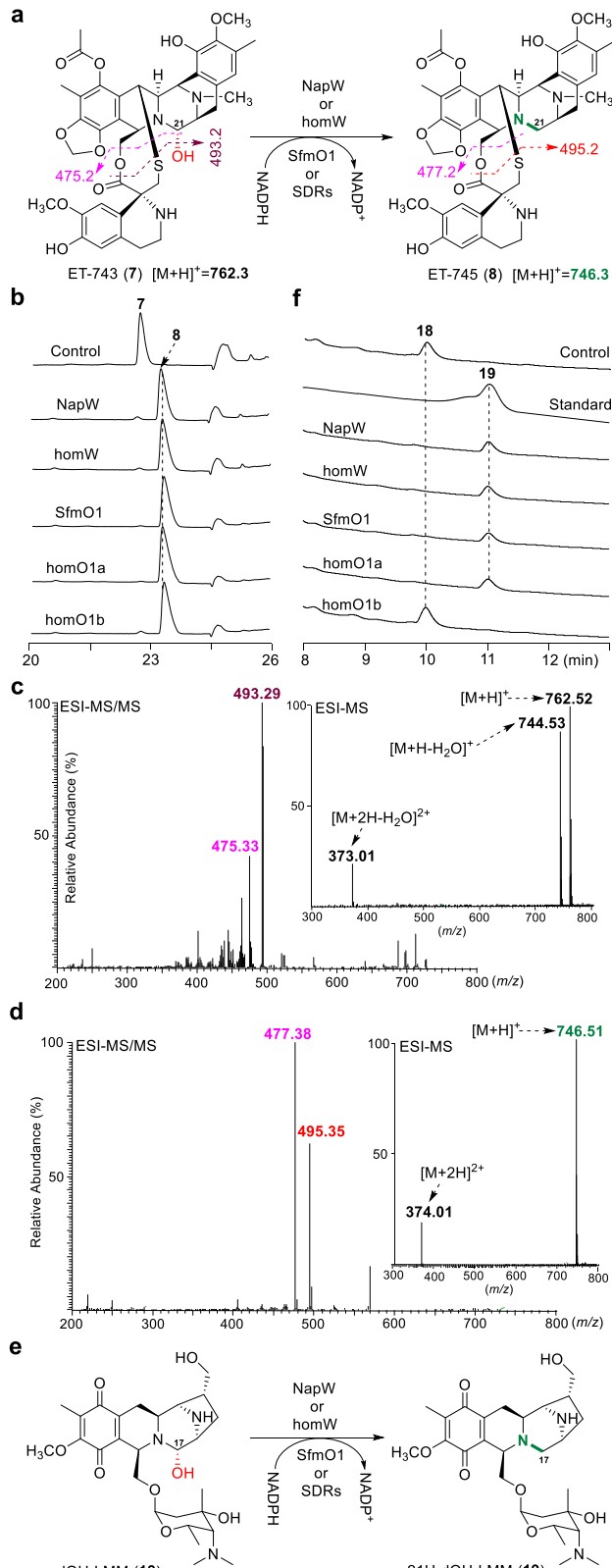

**Fig. 6 Biochemical assay of SDRs-catalyzed reduction using ET-743 and dOH-LMM as substrates.** SDRs-catalyzed reactions using **7** (**a**) or **18** (**e**) as substrate. HPLC analysis of enzymatic reactions with **7** (**b**) or **18** (**f**) as substrate. ESI-MS and ESI-MS/MS analysis of substrate **7** (**c**) and product **8** (**d**).

mechanisms are necessary but seem not completely competent to the task to guarantee the harmlessness during the whole biosynthetic process. Thereby another cryptic self-defense strategy was evolved by integrating with the pathway to ensure the survival of host during the biosynthetic process, which includes two key enzymatic steps via reduction-oxidation occurring at different timing and location (Fig. 7). In cytoplasm, SDRs (NapW/homW or SfmO1/homO1a/homO1b) will quench the reactive iminium intermediate immediately once it is generated by NRPS system via reduction of the C=N double bond to block the pharmacophore; which is a damage-control process because the resulted intermediates are totally harmless to the producer for deficiency of warhead. Subsequently, the matured but still inactivated intermediate **2** will be re-activated by secreted oxidase outside the host cells. Moreover, this secreted oxidase NapU could further oxidatively inactivate NDM into **3** to control the antibiotic concentration around the host cell (Fig. 1c). Despite this, the high active NDM still could enter the host cell and cause potential harm to the producer; these SDRs thereby set up the firewall by reductive inactivation of it via pharmacophore quenching (Fig. 7). This redox-governed cycle avoided the self-cytotoxicity and guaranteed no antibiotic consumption. Beyond that, the SDRs also confer the host cells general immunity toward most of the THIQ antibiotics because all the members of this family employ the common hemiaminal pharmacophore for DNA alkylation (Fig. 7). During this manuscript preparation, a fungal redox-mediated self-resistance cycle in macrolide A26771B biosynthesis was elucidated via reversible conversion of a ketone and alcohol[33]. While, compartmentalized biosynthesis of natural products is widely observed in eukaryotic cells of fungi to avoid self-harm[34,35]. Our discovery, featuring with enzymatic substrate promiscuity and widely distributed homolog genes, not only provided an omnibearing resistance mode toward THIQ family antibiotics in prokaryotic cells, but also displayed a multi-level and sophisticated damage-control in bacterial secondary metabolism.

One of the notable features of self-resistance in THIQ biosynthesis is taking advantage of reductive reaction to quench pharmacophore in situ apart from canonical modification on the end product. Actually, enzyme-guided chemical modification of compound is one of the most widely existed strategies for antibiotic resistance including self-resistance involved in biosynthetic pathways. However, these modifications are usually limited by reversible group-transfer reactions acting on end-products, such as methylation, acylation, phosphorylation, glycosylation, amino-, or peptide-acylation and so on[1,36]. Rare cases employing reductive reaction include virginiamycin M1 reductase-catalyzed conversion of a ketone to alcohol[37], a nitroreductase-mediated chloramphenicol inactivation and CytA-catalyzed dehydroxylation of cytorhodin[38,39], as well as two recently-discovered fungal examples including a multistep reduction cascade in monasone naphthoquinone and a redox-cycle in A26771B biosynthesis[33,40]. Most of these modifications change the structure of antibiotics to prevent them from binding to the responding cellular target inside the cell rather than destroy the pharmacophore. This discovery of SDRs, functional as imine reductase, mediated reductive inactivation of hemiaminal pharmacophore without destruction of the THIQ skeleton structure, thereby added another enzymatic reaction in bacteria beyond the typical hydrolysis for warhead inactivation.

Another significance of this self-resistance model in THIQ biosynthesis is the cryptically temporal and spatial shielding mode comprehensively integrated into the biosynthetic pathway. Antibiotic-producer usually evolved self-resistance strategy to deal with the final active compound; sometimes, analogs with similar structure could also be detoxified. Indeed, the SDRs encoded by the

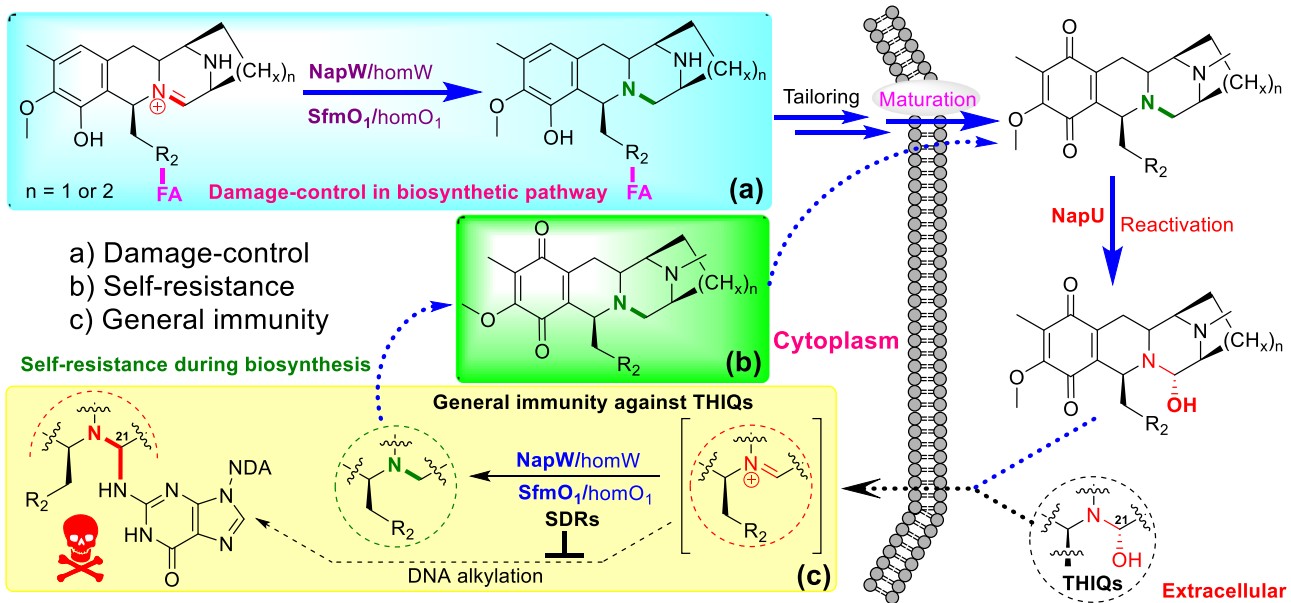

**Fig. 7 Summary of the multi-level resistance model involved in NDM and SFM biosynthesis. a** Damage-control process in secondary metabolic pathway controls the efficiency of biosynthetic machinery. **b** Self-resistance during biosynthesis by NapW-homologs ensures the safety of antibiotic-producing strains. **c** General immunity against THIQs is not only involved in antibiotic-producing hosts, but also exists in non-antibiotic-producing microbes containing NapW-homologs.

BGC and genome exhibit effectively reductive inactivation of THIQ antibiotics including NDM, SFM, ET-743, and LMM, which bear structural diversity despite belonging to the same family. Beyond that, enough amount of SDRs encoded by more than one copy of gene could also recognize the propeptide-linked intermediate, which led to completely blocking the pharmacophore and fundamentally destroying the warhead accompanied by the scaffold-formation. This modification step likely occurs at the early stage before tailoring modifications thereby resulting in the harmlessness of all the following metabolites in cytoplasm. Moreover, this reductive modification is also chemically favorable due to the improved stability of the resulted intermediate, which facilitates the biosynthetic steps. Thus, these SDRs not only confer general immunity toward natural occurring final THIQ antibiotics, but also play an important role in damage-control process during the entire biosynthetic route (Fig. 7). In recent years, more and more enzymes of unknown function are discovered to be functional as damage-control enzymes for repairing or inactivating the offending metabolites in primary metabolism[41]. Herein, the SDRs-catalyzed reduction in THIQ biosynthesis from *Steptomyces* represents a sophisticated damage-control in secondary metabolic pathway to accurately control the efficiency of biosynthetic machinery.

Extensive study of the molecular mechanisms of self-resistance, resistance/resistome are becoming a fundamental paradigm for combating the antibiotic resistance crises[42–44]. Based on the BGC of high potent secondary metabolites, systemic mining and deeply understanding the self-resistance determinants in the producers thereby provided a well-evolved model system for finding different resistance elements toward specific family of antibiotics[42]. Significantly, the homologous resistance genes could be wide-spread in nature including in the human microbiome, such accumulating evidences including NapW-like SDRs-mediated hemiaminal pharmacophore detoxification in THIQs reported here, GyrI-like cyclopropane hydrolase-catalyzed cyclopropyl warhead opening for YTM/CC-1065 pathway[9], and TnmS1-like sequestration proteins-guided enediyne resistance for tiancimycin biosynthesis[45]. With the increasingly and frequently clinical application of this family of antibiotics, the discovery of widespread NapW-like SDRs-mediated resistances may be critical for dealing with the potential clinical resistance problem.

In summary, we have identified an unprecedented self-resistance system in *Streptomyces* based on the elucidation of NDM biosynthetic pathway and verified by SFM-A biosynthesis. Apart from the common antibiotic effluxion by transporters and DNA-damage repair by UV-repair proteins, this sophisticated self-resistance machinery includes (i) propeptide formed by NRPSs, (ii) reductive inactivation of hemiaminal pharmacophore by SDRs in situ, (iii) prodrug maturation with inactivated intermediate exclusion, (iv) extracellularly oxidative reactivation by secreted oxidoreductase, (v) extracellularly over-oxidative inactivation to control antibiotic concentration, and (vi) intracellularly reductive inactivation of antibiotic/analogs. Considering the clinical usage of ET-743 and the potency of other members of THIQs[12,14,29], our findings thereby may be used to predict, determine the potent drug resistance and even inform the design of therapeutic agents that might not be subject to, or can circumvent further drug resistance in clinical settings. In addition, our enzymological studies show this family of SDRs functional as stereoselective imine reductase exhibiting excellent substrate-tolerance toward complex structure, thus provided more choices as biocatalyst for preparation of diastereomerically enriched complex saturated amine heterocycles[20,46,47].

## Methods

**Materials**. Biochemicals and media were purchased from Sinopharm Chemical Reagent Co. Ltd. (China); Oxoid Ltd. (UK); Sigma-Aldrich Corporation (USA); Shanghai Sangon Biotech Co. Ltd. (China) or Alibaba (China) unless otherwise stated. The enzymes for genetic manipulation were purchased from Thermo Fisher Scientific Co. Ltd. (USA); Takara Biotechnology Co. Ltd. (China) or New England Biolabs (USA). The primers and synthetic genes were synthesized by GENEWIZ Co. Ltd. (China). Chemical reagents were purchased from Sigma-Aldrich Corporation (USA), Shanghai Aladdin Bio-Chem Technology Co. Ltd. (China). All the primers, strains, and plasmids used are in Supplementary Table 1 and Table 2, respectively.

**Genome sequencing and analysis**. 100 μL of a spore solution of *S. lusitanus* NRRL 8034 was added to 50 mL of TSB medium. After incubation for 60 h at 30 °C with shaking at 220 rpm in a 250 mL flask equipped with sterile glass beads, the mycelia were collected by centrifugation at 4000 × *g* from 12.5 mL of cell suspension. Pellets

were resuspended in 10 mL of STE buffer (15% (w/v) sucrose, 25 mM EDTA, 25 mM Tris-HCl pH 8.0) twice, then a lysozyme solution was added to 5 mg/mL. After lysis at 30 °C for 15 min, 0.1 mL of STE buffer containing 10 mg/mL Protein K and 1 mL of a 10% (w/v) SDS solution were added and then mixed slowly by inversion and incubated for 15 min at 70 °C. 2.5 mL 5 M KAc was added and cooled down on ice for 15 min. A mixed solution of 5 mL saturated phenol (pH 7.9 ± 0.2) and 5 mL 24:1 (v/v) chloroform/isopropanol was added and mixed thoroughly by inversion to precipitate proteins. Two phases of the solution were separated by centrifuging for 15 min at 12,000 × g. The upper aqueous phase was transferred to a new tube and one equivalent 24:1 (v/v) chloroform/isopropanol was added and mixed for removing phenol. The upper aqueous phase was transferred to a new tube and two equivalent ethanol was added to precipitate DNA. The liquid was removed and 70% (v/v) of ethanol was added to wash the DNA before dissolving the resulting pellet in TE buffer (pH 8.0). Sequencing was performed at the Beijing Genomics Institute (Shenzhen, China). Gene analysis and functional annotation were performed by BioEdit, combined with 2ndFind and BlastP.

**Construction of inactivation mutants of *S. lusitanus* NRRL 8034.** The genes *napW*, *homW*, and dual genes were inactivated by in-frame deletion. To inactivate *napW*, a 2.1 kb *Eco*RI/*Xba*I fragment (amplified with primers: *napW*-L-for/rev, Supplementary Table 2) and a 3.92 kb *Xba*I/*Hin*dIII fragment (amplified with primers: *napW*-R-for/rev, Supplementary Table 2) were successively cloned from genome DNA of *S. lusitanus* NRRL 8034 and cloned into the *Eco*RI and *Hin*dIII sites of pKC1139, giving the recombinant plasmid pTG3032 in which 936 bp in-frame coding region of *napW* was deleted. The plasmid was then introduced into *S. lusitanus* NRRL 8034 by intergeneric conjugation from *E. coli* S17-1. The colonies that were apramycin-resistant at 37 °C were identified as the integrating mutants, in which a single-crossover homologous recombination event took place. These mutants were cultured for five rounds in the absence of apramycin. The resulting isolates that were apramycin-sensitive were subjected to PCR amplification to examine the genotype of the double-crossover mutant strain, as judged by a 634 bp desired product using the primers *napW*-gt-for and *napW*-gt-rev (Supplementary Fig. 2). Further sequencing of this PCR product confirmed the genotype of TG3022, in which gene *napW* was deleted in-frame. The inactivation of gene *homW* and dual genes were constructed in the similar way.

**Protein expression and purification.** DNA isolation and manipulation in *E. coli* or *Streptomyces* strains were performed following standard methods[48]. PCR amplifications were carried out on a Thermal Cycler (Applied Biosystems). The PCR products were cloned into the pMD19-T vector cloning kit (Takara Biotechnology (Dalian) Co. Ltd), confirmed by DNA sequencing (GENEWIZ Co., Ltd. China), digested with *Nde*I and *Hin*dIII, and ligated into the final expression vector pET-28a (+) and pET-37b (+), respectively.

The resulting recombinant plasmids were introduced into *E. coli* BL21(DE3) for protein overexpression. When the cultures (1 L) in LB media supplemented with kanamycin (50 μg/mL) were grown to an $OD_{600}$ of 0.6–1.0 at 37 °C, protein expression was induced by the addition of isopropyl-β-D-thiogalactopyranoside (IPTG) with a final concentration of 0.1 mM, followed by further incubation for 18–24 h at 16 °C. Then, the cultures were centrifuged for 10 min at 5000 × g 4 °C. The *E. coli* cell pellet was resuspended in ~30 mL of lysis buffer (50 mM Tris-HCl, 500 mM NaCl, 25 mM imidazole, and 10% glycerol, pH 8.0), and the His-tagged proteins were purified by Ni-NTA affinity chromatography according to the manufacturer's manual (Qiagen). The eluted protein was desalted using a PD-10 Desalting Column (GE Healthcare, USA) and the purified protein was stored at −80 °C in buffer (50 mM Tris-HCl, 200 mM NaCl, and 10% glycerol, pH 8.0) for further enzymatic assays. The concentration of the purified proteins was detected by DS-11+ spectrophotometer (DeNovix, USA).

**Site-directed mutagenesis.** PCR amplifications (18–26 cycles) were carried out using *napW* expressing plasmids as template. After gel extraction, the PCR products were digested by *Dpn*I and the digested products were directly transferred into *E. coli* DH5α. Each point mutation was confirmed by DNA sequencing. The resulting site-mutated plasmids were expressed in *E. coli* BL21 (DE3) according to the procedures described above for the WT protein expression.

**Bacterial growth inhibition assays.** The *E. coli* BL21(DE3) strains transformed with recombinant plasmids were induced for expression of NapW and homW for about 5 h at 16 °C. The resulting cultures were adjusted to the same concentration, and then transferred to fresh LB media in test tube containing IPTG (0.1 mM), kanamycin (50 μg/mL), and NDM with different concentrations to grow for about 12–24 h, respectively. The final *E. coli* cultures were diluted and the concentrations were measured by the absorbance at 600 nm, respectively. When the mutants and wild type of *S. lusitanus* NRRL 8034 were used as test strains, 2 μL different concentrations of NDM were dropped on the plate surface for 3–5 days of further incubation.

**Metabolite analysis.** High-performance liquid chromatography (HPLC) analysis was conducted on the Agilent 1200 HPLC system (Agilent Technologies Inc., USA) or Thermo Scientific Dionex Ultimate 3000 (Thermo Fisher Scientific Inc., USA)

with a reverse-phase Spursil C18 column (5 μ, 4.8 × 250 mm). Semi-preparative HPLC was performed on a Shimadzu LC-20-AT system. HPLC electrospray ionization MS (HPLC-ESI-MS) was performed on the Thermo Fisher LTQ Fleet ESI-MS spectrometer (Thermo Fisher Scientific Inc., USA). High-resolution ESI-MS analysis was conducted on the 6230B Accurate Mass TOF LC/MS System (Agilent Technologies Inc., USA). NMR data were collected on the Agilent 500 MHz NMR spectrometer (Agilent Technologies Inc., USA) and Bruker AVANCE III HD 600 spectrometers (Bruker Daltonics Inc., USA).

**Isolation and purification of compound 18.** *S. candidus* LL-AP191 (NRRL 3110) was cultured in 100 mL liquid medium (1 g soybean meal, 2 g glucose, 1 g corn steep liquor, 0.3 g $CaCO_3$) at 28 °C for 2 days. For fermentation, 3 mL inoculums were seeded in a 500 mL flask containing 100 mL of the fermentation medium (0.5 g peptone, 1.0 g glucose, 4 g molasses, 0.5 g $CaCO_3$) and incubated at 28 °C for 2–3 days. The supernatant was extracted twice with ethyl acetate at pH 9.0 and the extract was evaporated to remove ethyl acetate. The crude extracts of fermentation were subjected to the reverse-phase silica gel column chromatography. All the fractions were monitored by HPLC. The desired fractions were further purified by semi-preparative HPLC and lyophilized to obtain **18** with yellow-green color. The yield is 0.5 mg/L.

**Enzymatic assays of mutants and wild type of NapW to NDM.** Compound **2** was used as substrate, which underwent the first step of oxidative reaction catalyzed by NapU to give NDM[17]. The solution was filtered by a centrifugal filter unit to remove NapU, then 1.5 mM NADPH and mutants or wild type of NapW (~8.3 μM) were added into the filtrates containing 1 mM NDM. The reactions were incubated for 30 min at 30 °C, then quenched by the addition of equivalent methanol to remove proteins and detected by HPLC with gradients of acetonitrile with 1‰ formic acid (phase B) in water with 1‰ formic acid (phase A) at a flow rate of 1.0 mL/min: 0–5 min, 5–10% phase B; 5–10 min, 10–20% phase B; 10–15 min, 20–25% phase B; 15–20 min, 25–55% phase B; 20–22 min, 55–65% phase B; 22–23 min, 65–95% phase B; 23–25 min, 95% phase B; 25–27 min, 95–5% phase B; 27–30 min, 5% phase B; UV detection at 270 nm. Control reaction lacked the enzyme NapW.

**Enzymatic assays of NapW and homologous proteins to NDM.** The reaction solution contained 1 mM NDM, 2 mM NADPH, 50 mM Tris-HCl pH 8.0, and corresponding homologous proteins of NapW. The reactions were incubated at 30 °C for 3 h, then quenched by the addition of equivalent methanol to remove proteins and detected by HPLC.

**Enzymatic assays of NapW and homologous proteins to compound 16.** Dual Pictet-Spengler reactions catalyzed SfmC[28] were carried out in 50 mM HEPES pH 7.0 buffer containing 1 mM 3-OH-5-Me-*O*-Me-Tyr, 0.5 mM Myristoyl-Ala-glycinaldehyde, 1 mM ATP, 10 mM $MgCl_2$, 10 μM $MnCl_2$, 1 mM NADH, and 10 μM SfmC at 30 °C for 2 h. Then 2 mM NADPH and corresponding homologous proteins of NapW were added and the reactions were carried at 30 °C for 3 h, which were quenched by the addition of equivalent methanol to remove proteins and detected by HPLC.

**Enzymatic assays of NapW and homologous proteins to ET-743.** The reaction solution contained 290 μM ET-743, 1 mM NADPH, 50 mM Tris-HCl pH 8.0, and corresponding homologous proteins of NapW. The reactions were incubated at 30 °C for 8 h, then quenched by the addition of equivalent methanol to remove proteins and detected by HPLC.

**Enzymatic assays of NapW and homologous proteins to 18.** The reaction solution contained 112.5 μM compound **18**, 2 mM NADPH, 50 mM Tris-HCl pH 8.0 and corresponding homologous proteins of NapW. The reactions were incubated at 30 °C for 8 h, then quenched by the addition of equivalent methanol to remove proteins and detected by HPLC.

**Preparation of compound $^2$H-2.** 500 μL filtrate catalyzed by NapU contained 1 mM NDM, and 50 mM Tris-HCl pH 8.0. 2 mM $NADP^+$, 4 mM [1-$^2$H] D-glucose and 100 μM recombinant BmGDH and 51.5 μM NapW were added and the solution was incubated at 30 °C for 2 h to give $^2$H-2. 25 tubes of tandem enzymatic reaction solution were extracted twice with ethyl acetate and the extract was evaporated to remove ethyl acetate. The extract was subjected to the silica gel column chromatography and eluted with 10:1 $CH_2Cl_2$/$CH_3OH$ to yield <1 mg $^2$H-2, which was characterized by $^1$H NMR and $^2$H NMR spectra.

**Preparation of compound 19.** 100 mL solution containing 5 mg compound **18**, 4 mM NADPH, and 186 μM NapW was incubated at 30 °C for 24 h and monitored by HPLC until when **18** was completely converted to **19**. The solution was extracted twice with ethyl acetate at pH 9.0 and the extract was evaporated to remove ethyl acetate. The crude extract was further purified by reverse-phase silica gel column chromatography to yield 1 mg **19** with yellow-green color.

**Crystallization and structural determination.** The NapW was purified by Ni-NTA affinity column and then applied to a size exclusion column (Superdex 200 16/600, GE Health Care). Initial crystallization of NapW and NapW·NADPH complex were performed at 16 °C using the sitting-drop vapor-diffusion method and commercial crystallization kits in 2 μL drops containing an 1:1 mixture of the protein solution (25 mM Tris-HCl, pH 8.0, 50 mM NaCl) and a reservoir. To optimize NapW crystallizing condition, 2 μL NapW protein (10 mg/mL) was mixed with an equal volume of reservoir solution (1.2 M $(NH_4)_2SO_4$, 0.1 M BIS-TRIS propane pH 6.5) and 0.4 μL additive (0.1 M Betaine hydrochloride). Prior to the diffraction experiments, 30% glycerol was added as the cryo-protectant. For the NapW·NADPH complex, native NapW protein (15 mg/mL in 25 mM Tris-HCl, pH 8.0, 50 mM NaCl) was incubated with 5 mM NADPH at 30 °C for 30 min. Crystal of the NapW-NADPH complex was harvested under the crystallization condition with 1.5 M $(NH_4)_2SO_4$, 0.1 M BIS-TRIS propane pH 7.0, and cryo-protection with 30% glycerol. All X-ray diffraction data were recorded at the Shanghai Synchrotron Radiation Facility (SSRF). Data reduction and intergration was achieved with HKL2000 or HKL3000 package[49,50]. The structure of NapW was determined by molecular replacement using 2QQ5 as search model and NapW·-NADPH complex was solved by molecular replacement using NapW structure. Interactive cycles of model rebuilding and refinement were carried out using COOT, Phenix[51,52]. The overall quality of the structural models was checked by PROCHECK and MolProbity[53,54]. All statistics for data collection and structural refinement were listed in Supplementary Table 3. Structure figures were made using PyMOL[55].

**Umbrella sampling.** NapW·NADPH·NDM complex was obtained by molecular docking via AutoDock4.2[56]. To enhance sampling of the valuable conformations of NDM dehydration and reduction systems, umbrella sampling was employed. The distance d(OD2-O3) between the OD2 of Asp165 and O3 of C-7 hydroxy in NDM, and distance d(C7-C4N) between the C7 of NDM and the C4N of NADPH were used as the reaction coordinates. Taking 0.05 Å as step length, the former was scanned from 6.0 to 4.5 Å and the latter was scanned from 7.0 to 4.0 Å by applying a harmonic force constant of 200 kcal/(mol·Å²), respectively, and 0.1 ns MD simulations were carried out in each window. After that, the potential energies of mean force of the systems were computed via the weighted histogram analysis method (WHAM)[57]. According to the free energy surface, the structure with minimal free energy was found out.

**Molecular dynamic (MD) simulations.** Based on the structure with minimal free energy in umbrella sampling, the classical MD simulations were performed using the AMBER14 program[58] suite with ff14SB force field. The parameter preparation for NDM and NADPH was performed by Antechamber package. The partial atomic charges and missing parameters for the substrates were obtained from the RESP model[59] after structure optimization at the level of HF/6–31G*. All ionizable side chains were maintained in their standard protonation states at pH 7.0. The dehydration and reduction systems were immersed in an octahedral box of TIP3P water box and were neutralized with $Na^+$ using the AMBER14 LEAP module. Then proper minimizations were carried out to remove atomic collisions. After heating from 0 to 300 K in 50 ps, the systems were equilibrated for 50 ps to obtain a reasonable initial structure. Four 50 ns trajectories were obtained for each model based on the equilibrated structure, and two of them were extended to 150 ns to confirm the stability of key distances. The following analyses were also carried out via the AMBER14 program suite.

**Quantum mechanical/molecular mechanical (QM/MM) calculations.** In order to study the catalytic mechanism of NapW, QM/MM calculations were carried out via Gaussian 09 program[60]. The initial structure was originated from the representative structures of two extended 150 ns MD simulations. The computational model consisted of NADPH, substrate 1, Asp165, and other residues within 5 Å of NADPH and 1, 1202 atoms in total. The QM region included 1, the truncated part of NADPH, Asp165, and a water molecule. At the level of ONIOM (M062X[61]/6–31G*: Amber), the optimizations of minimums, transition states, and intrinsic reaction coordinate (IRC) were calculated. The transition state was confirmed by a single imaginary frequency and the correct vibrational vector. After that, M062X/6–311 + G** level was used to calculate the single point energy to improve the computational accuracy.

**Statistics and reproducibility.** Biological fermentation experiments were performed in at least triplicates. All enzyme assays and protein analysis experiments were verified with at least two independent enzyme preparations.

**Reporting summary.** Further information on research design is available in the Nature Research Reporting Summary linked to this article.

## Data availability

Data supporting the findings of this work are available within the paper and its Supplementary Information file, which includes six tables (Supplementary Tables 1–6) and 26 figures (Supplementary Figs. 1–26). A reporting summary for this article is

available as a Supplementary Information file. The genes homW, homO1a, and homO1b generated in this study have been deposited in GenBank under the accession code MT230905, MT230906, and MT230907. Structure data of NapW and NapW-NADP generated in this study have been deposited in the Protein Data Bank with the accession codes 7BTM and 7BSX. The raw data used for Supplementary Figs. 1, 2, 3, 6, 14, and 22 are provided in the Source Data file. All other data that support the findings of this study are accessible in the manuscript and the Supplementary Information. Source data are provided with this paper.

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

## Acknowledgements

This work was supported in part by grants from National Natural Science Foundation of China (31930002 and 21621002) and Chinese Academy of Sciences (QYZDJ-SSW-SLH037). The authors thank the staff of beamlines BL17U1 and BL18U1 of Shanghai Synchrotron Radiation Facility for access and help with the X-ray data collection. We appreciate Prof. Dawei Ma of SIOC for kindly providing the synthesized ET-743.

## Author contributions

G.-L.T. conceived the study and designed the experiments. W.-H.W., Y.Z., C.P., and J.-Y.P. participated in the genetic experiments; W.-H.W., Y.Z., and Y.-Y.Z. performed the biochemical assays; W.-H.W. and M.-C.T. carried out the chemically synthetic experiments; Q.Y., C.-C.J., Y.-L.Z., and T.S. provided the molecular dynamic simulations; W.-H.W., L.W., and J.Z. performed the protein crystal structure analysis. All authors analyzed and discussed the results; W.-H.W. and G.-L.T. wrote the manuscript.

## Competing interests

The authors declare no competing interests.
