## [Peer Review File · Nature Communications]

REVIEWER COMMENTS

Reviewer #1 (Remarks to the Author):

This manuscript reported a unique self-resistance strategy in the biosynthesis of tetrahydroisoquinoline (THIQ) antibiotics, which featured with reductive inactivation of hemiaminal pharmacophore by two SDRs, NapU and HomW. The functions of NapU and HomW were fully characterized by gene deletion, in vitro biochemical assay, as well as antibacterial activity comparison. The catalytic mechanism was also investigated by determining the crystal structure of NapW-NADPH complex and molecular dynamics simulation analysis. Interestingly, this detoxification strategy was also identified in the biosynthesis of other THIQs and similar SDRs were found in many bacteria and functioned for reductive inactivation of hemiaminal pharmacophore. The findings presented in this manuscript gave a full story for multi-dimensional self-resistance mode during THIQs biosynthesis, and exhibit a sophisticated damage-control in secondary metabolism and general immunity toward this family of antibiotics. This report on a unique SDR-mediated self-protection mechanism is of broad interest and a large amount of work was presented in the manuscript.

Overall, I really like this wonderful work and there are excellent interesting ideas. I think it's recommended to be published after minor revision.

1. Please updated the situation of Ref 31 and 34;
2. Page 8, line 203, "confirmed by MS" could change to "confirmed by HR-MS/MS";
3. Since hydrolysis reaction is not the only strategy for modification reactive pharmacophore, I think it is not necessary to emphasis its importance in pharmacophore modification. Such as the following sentence "While other types of enzyme-reactions beyond hydrolysis are little known for inactivation of the pharmacophore rather than destruction of the skeleton structure." may not suitable.
4. Do NapW-homologues exist in any pathogenic bacteria? This may lead to resistance to THIQ antibiotics and worth to discuss.
5. Please double check the coupling constants of protons showed in Tables S4-S5. The coupling constants should be consistent for direct coupling protons.
6. Please indicate the UV detection wavelength in figures (such as Figure S7, S12, S23-S26) relative to HPLC traces.

Reviewer #2 (Remarks to the Author):

Wen et al. investigated the functions of short-chain dehydrogenases during the biosynthesis of tetrahydroisoquinoline-type antibiotics. Gene deletion of previously uncharacterized genes napW and homW abolished the production of the natural product suggesting their roles in the biosynthesis. They showed by invitro assays that NapW and HomW catalyze reduction of hemiaminal intermediates that have been believed to be toxic to the producing organism and proposed that the

reductases are responsible for the drug resistance. The mechanism of the reduction reaction was also investigated by crystallographic structure analysis and labeling experiments. Based on the in vitro experiment, it was proposed that the reduction of the toxic imine/hemiaminal intermediates of tetrahydroisoquinoline is widely utilized by both tetrahydroisoquinoline-producing and non-producing microbes for resistance. This manuscript is very easy to follow, and the data presentation is excellent. The work expands our chemical knowledge regarding how microorganism produce toxic natural product while avoiding killing themselves. I believe that the work contributes to the field of natural product chemistry. However, I listed several concerns, and they should be addressed before publication.

1. The authors clearly demonstrated that some SDRs from non-antibiotic-producing microbes can reduce THIQs. However, whether those proteins only accept THIQ type substrates is unknown considering that the substrate specificity of SDRs is often compromised. Please make a comment on this point.

2. The reduction/oxidation step of THIQ is essentially reversible. It might be possible that the distribution of the redox pair of THIQ is thermodynamically controlled by the intra- or extracellular concentrations of the redox cosubstrates such as NADP⁺/NADPH. Can you briefly explain why THIQ is reduced in the cells and oxidized when it is exported?

3. It would be nice to show sequence alignment of NapW and homologues although the authors show an SSN figure in the SI. Is Asp165 in napW conserved among the homologues. If so, a similar mutation experiment targeting Asp165 should be performed using the homologues. Any other conserved residue/motif/domain that can explain selectivity for THIQ hemiaminal reduction?

4. As for the mechanism of NapW, it is likely that the tertiary amine is protonated in solution and the elimination of the water may occur in concerted with deprotonation from N. Is it possible that Asp165 serve as a base for deprotonation to promote the elimination. I feel that the function of Asp165 is not fully supported by the data. Soaking with the substrate and/or docking simulation would be helpful to get more insight.

5. In the discussion section, the description about the resistance mechanisms by efflux or DNA repair need citation of references.

6. Figure 3a: This is hard to see. The resolution and/or size should be increased.

Reviewer #3 (Remarks to the Author):

The authors have presented some novel functions of SDR enzymes, including structural and functional characterisation. Overall, this is interesting and will be an important addition to the literature. However, the manuscript does not convey the significance of why this work is particularly noteworthy. Many SDR's have unknown functions, and I don't believe it's an overly significant result that these SDR's are able to act on the compounds described.

AUTHOR'S RESPONSE TO REVIEWERS

Reviewer #1 (Remarks to the Author):

This manuscript reported a unique self-resistance strategy in the biosynthesis of tetrahydroisoquinoline (THIQ) antibiotics, which featured with reductive inactivation of hemiaminal pharmacophore by two SDRs, NapU and HomW. The functions of NapU and HomW were fully characterized by gene deletion, in vitro biochemical assay, as well as antibacterial activity comparison. The catalytic mechanism was also investigated by determining the crystal structure of NapW-NADPH complex and molecular dynamics simulation analysis. Interestingly, this detoxification strategy was also identified in the biosynthesis of other THIQs and similar SDRs were found in many bacteria and functioned for reductive inactivation of hemiaminal pharmacophore. The findings presented in this manuscript gave a full story for multi-dimensional self-resistance mode during THIQs biosynthesis, and exhibit a sophisticated damage-control in secondary metabolism and general immunity toward this family of antibiotics. This report on a unique SDR-mediated self-protection mechanism is of broad interest and a large amount of work was presented in the manuscript. Overall, I really like this wonderful work and there are excellent interesting ideas. I think it's recommended to be published after minor revision.

1. Please updated the situation of Ref 31 and 34;

R: Thanks for your remark. We have updated the situation of Ref 31 and 34, which have been changed into Ref 33 and 36 for the addition of two references.

Ref 33: Zhang, Y., et al. Self-resistance in the biosynthesis of fungal macrolides involving cycles of extracellular oxidative activation and intracellular reductive inactivation. *Angew. Chem. Int. Ed.* 60, 6639-6645 (2021).

Ref 36: Zhang, Q., Chi, H.-T., Wu, L., Deng, Z. & Yu, Y. Two cryptic self-resistance mechanisms in *Streptomyces tenebrarius* reveal insights into the biosynthesis of apramycin. *Angew. Chem. Int. Ed.* 60, 8990-8996 (2021).

2. Page 8, line 203, “confirmed by MS” could change to “confirmed by HR-MS/MS”;

R: Thanks. Indeed, product 17 was characterized by HR-MS/MS and we have changed it.

3. Since hydrolysis reaction is not the only strategy for modification reactive pharmacophore, I think it is not necessary to emphasis its importance in pharmacophore modification. Such as the following sentence “While other types of enzyme-reactions beyond hydrolysis are little known for inactivation of the pharmacophore rather than destruction of the skeleton structure.” may not suitable.

R: The sentence is corrected as “Therefore, continuous efforts on elucidation of new self-resistance mechanisms based on natural product biosynthesis will enrich our knowledge about enzyme-catalyzed inactivation of antibiotics, which may include the novel enzyme-reactions acting on the pharmacophore beyond hydrolysis.” (shown in Page 3, Line 13-16.)

4. Do NapW-homologues exist in any pathogenic bacteria? This may lead to resistance to THIQ antibiotics and worth to discuss.

R: Yes, NapW-homologues exist in some pathogenic bacteria, such as SDR-Li (*Leptospira interrogans*, Spirochaetes), which mentioned and characterized in our manuscript. Short-chain dehydrogenases/reductases (SDR) form a large, functionally heterogeneous protein family and are widely distributed in numerous organisms more than pathogenic bacteria. Only a few of them have been functional characterized to perform other functions, not resistance to THIQ. The primary sequence homology between SDRs can be very low and we don't think all SDRs in any pathogenic bacteria can defend against THIQ antibiotics.

5. Please double check the coupling constants of protons showed in Tables S4-S5. The coupling constants should be consistent for direct coupling protons.

R: Thanks for your remark. We have checked the coupling constants of protons in NMR spectroscopic data of compound **18** and **19**. The coupling constants should be same for direct coupling protons theoretically. Sometimes there is a subtle difference between them. For compound **18**, the coupling constants of 2'-CH₂ are 14.9 Hz and 14.8 Hz; the coupling constants of 5'-CH and 6'-CH₃ are 7.3 Hz and 7.2 Hz; the coupling constants of 13-CH and 17-CH are 3.0 Hz and 3.2 Hz. Similar situation appears in compound **19**. These are within the margin of error.

6. Please indicate the UV detection wavelength in figures (such as Figure S7, S12, S23-S26) relative to HPLC traces.

R: Thank you for your suggestion. We have indicated the UV detection wavelength (270 nm) in the mentioned figures.

Reviewer #2 (Remarks to the Author):

Wen et al. investigated the functions of short-chain dehydrogenases during the biosynthesis of tetrahydroisoquinoline-type antibiotics. Gene deletion of previously uncharacterized genes napW and homW abolished the production of the natural product suggesting their roles in the biosynthesis. They showed by in vitro assays that NapW and HomW catalyze reduction of hemiaminal intermediates that have been believed to be toxic to the producing organism and proposed that the reductases are responsible for the drug resistance. The mechanism of the reduction reaction was also investigated by crystallographic structure analysis and labeling experiments. Based on the in vitro experiment, it was proposed that the reduction of the toxic imine/hemiaminal intermediates of tetrahydroisoquinoline is widely utilized by both tetrahydroisoquinoline-producing and non-producing microbes for resistance. This manuscript is very easy to follow, and the data presentation

is excellent. The work expands our chemical knowledge regarding how microorganism produce toxic natural product while avoiding killing themselves. I believe that the work contributes to the field of natural product chemistry. However, I listed several concerns, and they should be addressed before publication.

1. The authors clearly demonstrated that some SDRs from non-antibiotic-producing microbes can reduce THIQs. However, whether those proteins only accept THIQ type substrates is unknown considering that the substrate specificity of SDRs is often compromised. Please make a comment on this point.

R: Short-chain dehydrogenases/reductases (SDR) form a large, functionally heterogeneous protein family and are widely distributed in microorganisms. The function of them is largely unknown. In fact, we carried out the *in vitro* enzymatic assays to discover the phenomenon that some SDRs from non-antibiotic-producing microbes can reduce THIQs and proposed the wide-spread resistance for THIQ antibiotics. The compatibility of the SDRs for other unknown substrates need further detailed research about the non-antibiotic-producing microbes.

2. The reduction/oxidation step of THIQ is essentially reversible. It might be possible that the distribution of the redox pair of THIQ is thermodynamically controlled by the intra- or extracellular concentrations of the redox cosubstrates such as NADP⁺/NADPH. Can you briefly explain why THIQ is reduced in the cells and oxidized when it is exported?

R: The reduction and oxidation steps in NDM biosynthesis were catalyzed by different enzymes. NapW/homW-NADPH expressed intracellularly ensured reducing state of propeptide linked intermediate (Fig. 7 in manuscript, process a), which was hydrolyzed by a peptidase to give the inactive product **2**. The secreted flavoprotein NapU (using covalent FAD as cofactor but not NADP⁺) catalyzed oxidative activation of inactive compound **2** to afford toxic product NDM extracellularly (Reference 17 in manuscript). Despite NapU can reduce the concentration of NDM by overoxidation, the toxic product still exhibits the ability to diffuse into cells. Then NapW/homW-NADPH quenched the warhead of NDM for self-resistance by reduction again (Fig. 7 in manuscript, process c, b). In brief, NapU is mainly responsible for activation extracellularly and NapW/homW-NADPH ensure intracellular safety.

3. It would be nice to show sequence alignment of NapW and homologues although the authors show an SSN figure in the SI. Is Asp165 in napW conserved among the homologues. If so, a similar mutation experiment targeting Asp165 should be performed using the homologues. Any other conserved residue/motif/domain that can explain selectivity for THIQ hemiaminal reduction?

R: We thank the reviewer to raise this important question. Sequence alignment of NapW and homologues is listed below. Residue Asp165 labeled by yellow background is conserved in most of NapW-homologues except for SDR-Ss and SDR-Cs which have low sequence homology with NapW. The residue Tyr in the two SDRs replaces Asp, which didn't affect enzymatic activity for some THIQs. Meanwhile, our data showed mutation on Asp165 didn't completely deactivate NapW, suggesting that Asp165 may not be the exclusively key factor. Other THIQ substrates containing

more complex framework than NDM also can be accepted by NapW and homologues. We have tried to crystallize NapW-homologues and cocrystallize NapW•NADPH complex with other THIQ substrates for illustrating conserved residue/motif/domain, but failed. Therefore, we can't draw a conclusion about which conserved residue/motif/domain responsible for selectivity for THIQ hemiaminal reduction; we only conclude that Asp165 is an important residue for enzymatic activity of NapW.

Alignment of NapW- homologues by Clustal Omega.

SDR-Ss	-----MKTLLQGKVAIIAGASRGVGRGVALGLAEQAGATVYVVGR	38
SDR-Cs	-----MAALHGSAIVAGGSRGAGAVARALGSAGATVYVTGR	38
SDR-Li	-----MNQKNLEGKAALVTGATRGAGRGIALALGEAGVTYVYVGR	40
Sfm01	-----VTDGVRTLAGKVALVAGGTRGGGRIAVELGAAGATVYVSGR	42
SDR-Mb	-----MNEDRRPLEGKIALVAGATRGAGRGIAIELGAAGATVYCSGR	42
Hom01a	---VTGVTFTGVHREVMHKTPTPLTGRALVAGATRGAGRALAVELGRAGATVYVTGR	56
S1vW2	MDDMSNEDITRQTENTEQAGPKGPLAGRIALVAGATRGAGRAQAVELGRAGATVYVTGR	60
Hom01b	-----MTWSPDPDALGRVAVVAGATRGAGRGFAAALGEAGATVYVCTGR	44
SDR-Rm	-----MTAALSGKIALVAGGTRGAGRGIAVELGAAGATVYVTGR	39
NapW	-----MERTPDPAPDLRGKIALVAGATRGAGRAIIVQLGAAGATVYVTGR	46
HomW	-----MTQPLRDKVALVAGATRGAGRGIAAELGAAGATVYVTGR	39
S1vW1	-----MTGSSKGPLAGRVALVAGATRGAGRGIAVELGAAGATVYVTGR	43
SDR-Pb	-----	0
SDR-Cc	-----MNQALKNKVALVAGATRGAGRGIAVELGAAGATVYATGR	39
SDR-Ss	TEKDDELP-----QFLK-GTTIYQAVDKVNELGGTGIAVKCDLRKDEEVEKFFKT	87
SDR-Cs	TVRGGQP-----PLDGAAGTVDDTAAEVTRRGGCGIAAPCDHTDDAQVEGLFAR	87
SDR-Li	SIKDK-H-----SEMNL-KETIEETAERINKLGGKAIWAQVDHTNPKEVKS LFER	88
Sfm01	SSSANGS-----SDMGR-PETIEETARAVTAAGGVGIPVRTDHSSPEEVRALVDR	91
SDR-Mb	SSRADVAGRRAPDARPFELSGR-PETIEETAELVTAAGGTGIAMRTDHLDEDVAALVKR	101
Hom01a	TTRE--RV-----SEVGRTTETIEESAELVTAAGGTGIAVPTDHLDEEQVRLVSR	105
S1vW2	TTRA--RA-----SEVGRTTETIEETAELVTAAGGTGIAVPTDHLDEAQVRLVSR	109
Hom01b	SSVRGRDG-----SDYDR-PETIEDTADLVSRLLGGTGIAVQADHLDPEQVRRLAGR	94
SDR-Rm	TTRA--KQ-----SEYGR-PETIEETAEMVTAAGGKGIACVDHTVPAEVEALIGA	87
NapW	TTRE--RR-----SEYNR-SETIEETAELVTEAGGTGIAVPTDHLVPEQVRLADR	94
HomW	STRA--RR-----SEYDR-PETIEDTADLVTEAGGHGIAVPADHLDPAQVEALVDR	87
S1vW1	STRA--RR-----SEYDR-PETIEDTADLVTEAGGHGIVAVPADHLDPDQVAAVVDR	91
SDR-Pb	-----QVDHLQPEQVQAL IAR	16
SDR-Cc	TTHA--QR-----SEYNR-PETIEETAELVNQAGGHGIPVQVDHLDPTQVQALVAR	87
* * .		
SDR-Ss	VNR-EQGRLDILVNSAWPASDHIMKGYFSNTPFWEQPLSFF---DDFIRVGVRSNYVASR	143
SDR-Cs	VQQ-EQGRLDLLVNAVWGGNELPSLQADWGRPCWQQQGAAGWEAMFTAGV RPALIASY	146
SDR-Li	IDQ-EQGKLNILVNDIWGGDPF----IEWSCKFWEHSLENG---LKVQKNCLNSHLITNY	140
Sfm01	IDAEQQGRLDVLVNCVWGGDRL----TDWNRPLWQQDLDKG---LRLLRQAVETHVITSR	144

SDR-Mb IRD-EHGRLDVLVNDVWGGDAL----TEWGKPFWELDLEQG---RVLLDRAIRTHVVTSR 153
 Hom01a IDR-DHGRLDVLVNDLWGGEHLTAG-SVFGKKSWEPLADG---LRILELGVRSHVITAA 160
 S1vW2 IDR-EYERLDILVNDLWGGEHLLAT-SVFGKKSWEPLADG---LRILELGARSHVITAA 164
 Hom01b LRD-EYGHIDVLVNDIWGAELKGGPAQWNTPVWEHDLDDG---LRILRLAVDTHLITAH 150
 SDR-Rm IRH-EQDRLDILVNDIWGGEHL----TEWVKPVWEHSLHKG---LRMLRLAIDTHLITAH 139
 NapW VDT-EQGRLDVLVNDVWGGERL----FEFDKKVWEHDL DAG---LRLMRLGVDTHAISSH 146
 HomW IAA-EQGRLDVLVNDIWGGETL----FDWDAPVWEHDLKDG---LRLRLAVETHAITSH 139
 S1vW1 IAS-EQARLDILVNDIWGGETL----FEWDSVPWEHDLKDG---LRLRLAVETHAITSH 143
 SDR-Pb IEK-EQGRLDVLVNDVWGAENL----ADWNPVWDHSLERG---FRMLRLGIDTHLITSH 68
 SDR-Cc IDS-EQGRLDILVNDI-GGEYL----AEFHQPVWQLSLEKG---LRLFRLAIDTHIVTSH 138

: : :::*** .. *: . ::

SDR-Ss LAAQMMTKQKS---GLIVNISYFA---GRRYWFNVANGVCKAAIDKLSADTAHELQYG 196
 SDR-Cs HAARLMAAQS---GLVVNVTYAFPGDKAGTYLGHLLYDLAKVSLSRFAFLAEELRPHG 203
 SDR-Li FATPLMIRNHS---DLILEITDGDY---RYRGSVYYTLVKYSIINLASSLSEELKPYG 193
 Sfm01 YAVRLMAARRS---GLVVEVTDGNTA---RYRGSFFYDVAKSTVIRLAFQAADLKEHG 197
 SDR-Mb HAVPLLLERRSLERRLIVEITDGDAM---YYRGNFFYDIAKTTVIRLAFAMSEELREHG 209
 Hom01a LALPLLIRSDA---PLHVEVTDGTARSNR-RYRENLYYDLAKNAPIRIAFGLGQELAEYG 216
 S1vW2 LLLPLLIRSDA---PLHVEVTDGTAHSNR-RYRENIYYDLAKNAPIRIAFGLAQELAEYE 220
 Hom01b HLLPLLIARPG---GLLVEVTDGTTGINASWYRISAFYDLAKAAVNRLAFSLGHELAPYG 207
 SDR-Rm FALALMIERP---GLLVELTDGTAEYNAAHYRLCPYYDLAKTGAIIRMAWAHAKDLALHG 196
 NapW FLLPLLVRPPG---GLVVEVTDGTAAYNGSHYRNSYFYDLVKNSVLRMGYVLAHELEPYG 203
 HomW HALPLLRRHPG---GLVVEVTDGTADYNRDHYRVSFFYDLAKSSVLRMAFALGHELGPRG 196
 S1vW1 HALPLLRRPPG---GLVVEVTDGTDAYNRDHYRNSFFYDLAKTSVLRMAFSLGHEVGPRG 200
 SDR-Pb FALPLLIRNKN---GLVVEVTDGTAEYNYKNYRISMFYDLVKNSVIRMAQSLAHELAPYQ 125
 SDR-Cc FALPLLIRNKN---GLVVEITDGTAEYNNKNYRSLFYDLAKTSVIRMAWALAQELKPHQ 195

:: * :::: * : * .. . ::

SDR-Ss VTVISLYPDTVRTEGMIALAHQDK-----SIDLSDSESPQFVGRCA 238
 SDR-Cs AAALALSPGHMRTERVLQHFQDEQHWQE-----EPGLAGSESPYELGRAVA 250
 SDR-Li VTVISLTPGFLRSEAMLDYFGVLEENWKDAILK-----DPHFIASETTAYIGRAVV 244
 Sfm01 VAAVAITPGFLRSEAMLEHFGVTEDNWRDGVAR-----DPDFAHSETPAYLGRAVA 248
 SDR-Mb VAAVAVTPGFLRSEAMLEHFGVTETWRDGAKK-----DPHFVIFSETPRFVGRGIA 260
 Hom01a GTAVAVTPGFLRSEQMLAHFGVTERNWRDAVAR-----EPHFAVAESPHYLARGVA 267
 S1vW2 GTAVAVSPGFLRSEQMLSHFGVSEENWRDAIAQ-----EPTFAIAESPHYLARTVA 271
 Hom01b ATAVAVTPGWL RSEMMLDNFGVTEATWREALAPGRAGGLPTAPEGFARSESPRYVGRAAA 267
 SDR-Rm ATSVALTPGWMRSEMMLDIFSVTEENWRDATSM-----QPHFVIFSETPRFTGRAVA 247
 NapW GTAVTLTPGWMRSEMMLLET LGVTEENWRDALTE-----VPHFCISESPSYVGRAVA 254
 HomW ATAVALTTPGWMRSEIMLDHFGVREDNWRDALEK-----VPHFAISETPRYVGRAVA 247
 S1vW1 ATAVALTTPGWL RSEIMLDHFGVREENWRDALDR-----VPHFAISETPRYVGRAVT 251
 SDR-Pb CTAVAMTPGWMRSEIMLDHFGVKEENWRDAAEK-----EPHFIISESPRYVGRAVA 176
 SDR-Cc CTAVALTTPGWL RSEIMLDHFEVSEANWQDATAK-----EPHFVIFSETPHYIGRAVA 246

: ::: * . :*: * : : : * : .

SDR-Ss	ALANDENI-MNETGKILITAEVAQHYGFTDVGRRPKSQREELW-----	281
SDR-Cs	ALAADPNV-LRRSGQVLLVGELARQYGFDTIDGTQPPPFVPL-----	292
SDR-Li	SLAEDPNV-FLKTGSATSTWRLSEEYNTDLTVVNHIGDNTSEKNLEKI--YKDMKQFFQ	301
Sfm01	ALAADPDI-MAKTGRALATWGLYKEYGFTDIDGSQPDFAAHWARTLEPRLGPLG-----	301
SDR-Mb	ALASDPEI-MRRSGGLFSSWQLAAEYIGDIDGTRPDWGSAAAGSSFAEEHRASHERFVH	319
Hom01a	ALAADPDRAARWNGRSVSSADLAGAYGFTDVGSRPDWAYFEDVWYGG-KDASPDDYR-	325
S1vW2	ALAADPDRAKRWNGKSTSSGELARAYGFTDVGSRPDWAYFEDVTYGG-KEASPDDYR-	329
Hom01b	ALAADPDR-ARWNQRSVTAELARVYGFDTIDGTRPDGWHDR-----	308
SDR-Rm	ALAADPER-ARWNGQSLSSGGLAKVYGFDDVVGSRPDCWRYMDEVMDMG-KAADATGYR-	304
NapW	ALAGDADV-ARWNGQSVSSGQLAQEYGFDTLDGSRPDCWRYLVEVQEAG-KPADPSGYR-	311
HomW	ALAGDPEV-ARFNGASLSSGGLAQEYGFDTLDGSRPDCWRYLVEVQDAG-RPADVTGYR-	304
S1vW1	ALAADPGV-ARFNGRSFSSGSLAREYGFDTLDGSRPDWAYRYLVEVQDAD-KPADVTGYR-	308
SDR-Pb	ALAGDPEA-ARWNGKSLSSGGLAKVYGFDTLDGSRPDCWRYLVEVQEAG-KPADASGYR-	233
SDR-Cc	CLAGDPDV-ARWNGQSLSSGGLAKVYGFDTLDGSRPDWAYRYICEVEAVG-KPADATGYR-	303

. ** * . : * . * : .

SDR-Ss	-----	281
SDR-Cs	-----	292
SDR-Li	SNKLEI-----	307
Sfm01	-TPL-----	304
SDR-Mb	GTTARHAVRAPDASGSV	336
Hom01a	-----	325
S1vW2	-----	329
Hom01b	-----	308
SDR-Rm	-----	304
NapW	-----	311
HomW	-----	304
S1vW1	-----	308
SDR-Pb	-----	233
SDR-Cc	-----	303

4. As for the mechanism of NapW, it is likely that the tertiary amine is protonated in solution and the elimination of the water may occur in concerted with deprotonation from N. Is it possible that Asp165 serve as a base for deprotonation to promote the elimination. I feel that the function of Asp165 is not fully supported by the data. Soaking with the substrate and/or docking simulation would be helpful to get more insight.

R: Thanks for your remark. Combined with NMR analysis of enzymatic product ²H-2, the direction of hydride supplied by NADPH is identical to that of hydroxyl leaving, which illustrates the reductive reaction is an S_N1 process containing two steps, but not an S_N2 process that means the elimination of the water occurs in concerted with the hydride delivery from NADPH. NADPH provided hydride (H⁻), not proton(H⁺). So Asp165 cannot serve as a base to accept H⁻.

We have tried hard to soak different substrates into the binary complex NapW•NADPH crystal for mechanism studies but all failed. We therefore have to carry out molecular dynamics simulation

experiments for explanation and identify Asp165 as an important residue for enzymatic activity of NapW.

5. In the discussion section, the description about the resistance mechanisms by efflux or DNA repair need citation of references.

R: Thanks for your advice. We have cited references (16,24,31 and 16,24,32) about the two resistance mechanisms.

6. Figure 3a: This is hard to see. The resolution and/or size should be increased.

R: Thank you for your advice. We have adjusted the resolution of Fig. 3a.

Reviewer #3 (Remarks to the Author):

The authors have presented some novel functions of SDR enzymes, including structural and functional characterization. Overall, this is interesting and will be an important addition to the literature. However, the manuscript does not convey the significance of why this work is particularly noteworthy. Many SDR's have unknown functions, and I don't believe it's an overly significant result that these SDR's are able to act on the compounds described.

R: Thanks for your remarks. Indeed, we don't think the true physiological function of all the characterized SDRs in their original hosts is to defend against THIQ antibiotics although they could reductive inactivation of THIQs. This reaction may be one of the side-reactions which conserved (or gained) during their evolution process.

For the significance of this work, we have discussed in detail in the "Discussion" part. Now we re-summarized in following:

1) we filled in the missing intracellular events of a multi-dimensional temporal-spatial shielding mode for the self-resistance during THIQs biosynthesis in *Streptomyces*. This sophisticated self-resistance machinery includes i) propeptide formed by NRPSs, **ii) reductive inactivation of hemiaminal pharmacophore by SDRs *in situ***, iii) prodrug maturation with inactivated intermediate exclusion, iv) extracellularly oxidative reactivation by secreted oxidoreductase, v) extracellularly over-oxidative inactivation to control antibiotic concentration, and **vi) intracellularly reductive inactivation of antibiotic/analogues**. Considering the clinical usage of ET-743 and the potency of other members of THIQs, our findings thereby may be used to predict, determine the potent drug resistance and even inform the design of novel therapeutic agents that might not be subject to, or can circumvent further drug resistance in clinical settings. Additionally, this discovery not only provided an omnibearing resistance mode toward THIQ family antibiotics in prokaryotic cells, but also displayed a multi-level and sophisticated damage-control in bacterial secondary metabolism.

2) SDRs performed reductive reaction to quench pharmacophore *in situ* is different from canonical modification on the end product. Previously, enzyme-guided chemical modifications of antibiotics are usually limited by reversible group-transfer reactions acting on end-products, such as methylation, acylation, phosphorylation, glycosylation, amino- or peptide-acylation and so on. Most of these modifications change the structure of antibiotics to prevent them from binding to the responding cellular target inside the cell rather than destroy the pharmacophore. This discovery of

SDRs, functional as imine reductase, mediated reductive inactivation of hemiaminal pharmacophore without destruction of the THIQ skeleton structure, thereby added another enzymatic reaction in bacteria beyond the typical hydrolysis for warhead inactivation.

3) These SDRs not only confer general immunity toward natural occurring final THIQ antibiotics, but also play an important role in damage-control process during the entire biosynthetic route. The SDRs encoded by the BGC and genome exhibit effectively reductive inactivation of THIQ antibiotics including NDM, SFM, ET-743, and LMM, which bear structural diversity despite belonging to the same family. Beyond that, enough amount of SDRs encoded by more than one copy of gene could also recognize the propeptide linked intermediate, which led to completely blocking the pharmacophore and fundamentally destroying the warhead accompanied by the scaffold-formation. This modification step likely occurs at the early stage before tailoring modifications thereby resulting in the harmlessness of all the following metabolites in cytoplasm. Moreover, this reductive modification is also chemically favorable due to the improved stability of the resulted intermediate, which facilitates the biosynthetic steps. Herein, the SDRs-catalyzed reduction in THIQ biosynthesis from *Streptomyces* represents a sophisticated damage-control in secondary metabolic pathway to accurately control the efficiency of biosynthetic machinery.

Sincerely yours,

Gong-Li Tang, PhD.
State Key Lab. of Bio-organic and Natural Products Chemistry
Shanghai Institute of Organic Chemistry, Chinese Academy of Sciences
345 Lingling Rd. Shanghai 200032, CHINA
Tel: +86-21-54925113, Fax: +86-21-64166128
E-mail: gltang@sioc.ac.cn

REVIEWERS' COMMENTS

Reviewer #1 (Remarks to the Author):

I believe the revised manuscript is acceptable for publication at Nat Comm.

Reviewer #2 (Remarks to the Author):

The authors have responded to reviewers' concerns appropriately, and the manuscript is now acceptable for publication. I do not have any more concerns.